# High Incidence of Amoxicillin-Induced Crystal Nephropathy in Patients Receiving High Dose of Intravenous Amoxicillin

**DOI:** 10.3390/jcm9072022

**Published:** 2020-06-27

**Authors:** Anne-Sophie Garnier, Juliette Dellamaggiore, Benoit Brilland, Laurence Lagarce, Pierre Abgueguen, Alain Furber, Erick Legrand, Jean-François Subra, Guillaume Drablier, Jean-François Augusto

**Affiliations:** 1Service de Néphrologie-Dialyse-Transplantation, Université d’Angers, CHU Angers, 49000 Angers, France; annesophie.garnier@chu-angers.fr (A.-S.G.); benoit.brilland@chu-angers.fr (B.B.); jfsubra@chu-angers.fr (J.-F.S.); 2Faculté de Pharmacie, Université d’Angers, 49000 Angers, France; judlmg6@gmail.com; 3Service de Pharmacologie-Toxicologie et Pharmacovigilance, Université d’Angers, CHU d’Angers, 49000 Angers, France; lalagarce@chu-angers.fr (L.L.); guillaume.drablier@chu-angers.fr (G.D.); 4Service de Maladies Infectieuses et Tropicales, Université d’Angers, CHU d’Angers, 49000 Angers, France; piabguegen@chu-angers.fr; 5UMR CNRS 6015–INSERM U1083, Équipe Physiopathologie Cardiovasculaire, UFR Santé, 49000 Angers, France; alfurber@chu-angers.fr; 6Service de Rhumatologie, Université d’Angers, CHU d’Angers, 49000 Angers, France; ErLegrand@chu-angers.fr; 7CRCINA, INSERM, Université de Nantes, 49000 Angers, France; 8Université d’Angers, 49000 Angers, France

**Keywords:** amoxicillin, crystalluria, acute kidney injury, incidence

## Abstract

Background: Amoxicillin (AMX)-induced crystal nephropathy (AICN) is considered as a rare complication of high dose intravenous (IV) AMX administration. However, recently, its incidence seems to be increasing based on French pharmacovigilance centers. Occurrence of AICN has been observed mainly with IV administration of AMX and mostly under doses over 8 g/day. Given that pharmacovigilance data are based on declaration, the real incidence of AICN may be underestimated. Thus, the primary objective of the present study was to determine the incidence of AICN in the current practice. Materials and Methods: We conducted a retrospective study between 1 January 2015 and 31 December 2017 in Angers University Hospital. Inclusion criteria were age over 18 years-old and IV AMX administration of at least 8 g/day for more than 24 h. Patients admitted directly into the intensive care units were excluded. Medical records of patients that developed Kidney Disease:Improving Global Outcome (KDIGO) stage 2–3 acute kidney injury (AKI) were reviewed by a nephrologist and a specialist in pharmacovigilance. AICN was retained if temporality analysis was conclusive, after exclusion of other causes of AKI, in absence of other nephrotoxic drug administration. Results: A total of 1303 patients received IV AMX for at least 24 h. Among them, 358 (27.5%) were exposed to AMX doses of at least 8 g/day and were included. Patients were predominantly males (68.2%) with a mean age of 69.1 years-old. AMX was administered for a medical reason in 78.5% of cases. Patients received a median dose of AMX of 12 g/day (152.0 mg/kg/day). Seventy-three patients (20.4%) developed AKI, 42 (56.8%) of which were KDIGO stage 2 or 3. Among the latter, AICN diagnosis was retained in 16 (38.1%) patients, representing an incidence of 4.47% of total patients exposed to high IV AMX doses. Only female gender was associated with an increased risk of AICN. AMX dose was not significantly associated with AICN development. Conclusion: This study suggests a high incidence of AICN in patients receiving high IV AMX doses, representing one third of AKI causes in our study. Female gender appeared as the sole risk factor for AICN in this study.

## 1. Introduction

Amoxicillin (AMX) is one of the most widely prescribed antibiotics used for antimicrobial prophylaxis or the treatment of established infections. Its main side effects are well known, however, in last years, AMX-induced crystal nephropathy (AICN), a complication thought to be rare, has been reported with a high incidence in France [1,2].

Nephrotoxicity of AMX is driven by two main mechanisms. The first is acute interstitial nephritis, a drug hypersensitivity reaction, which classically manifests after 7 to 10 days following drug exposition. In its classical form, patients present with acute kidney injury (AKI), back pain, fever, leukocyturia, eosinophiluria, microscopic hematuria, and extra-renal involvement such as arthralgias, skin rash, and liver test disturbance (cytolysis, cholestasis) [3]. The second way by which AMX may induce AKI is crystalluria. Kidney stones and crystalline nephropathy are frequent complications of drug administration [4]. Indeed, a multitude of drugs have been reported to induce crystalluria, among them, protease inhibitors used for human immunodeficiency virus treatment and sulfadiazine represent classical and frequent causes [5,6]. In crystalline nephropathy, kidney injury occurs as a consequence of crystal precipitation within tubules. Pathophysiology of crystal formation is related to two main mechanisms. The first one is the direct precipitation of the drug due to its poor solubility in urine. The second one is the crystallization of endogenous metabolic compounds because of the drug effects [5,6]. In this latter case, the drug is not found within the crystals (i.e., anhydrase inhibitors and calcium oxalate or calcium phosphate crystals). Several factors are well known to induce crystal precipitation in urine such as high urine drug excretion, high dose administration, and poor diuresis leading to high urinary drug concentration are the main risk factors [4].

The first case of AMX-induced crystalluria was reported in 1985 by Sjovall et al. in a heathy volunteer taking part in a study comparing the renal clearance of intravenous ampicillin and AMX [7]. The first case of AICN was reported in 1999 in a 4-year-old boy following AMX overdose [8]. Until recently, AICN was reported in the medical literature sporadically and mainly in single case reports [9,10,11,12]. In 2018, Vodovar et al. reported a cohort of 45 patients with AICN which were identified from the declarations done in Paris regional centers of pharmacovigilance (RPV) in France between 1985 and 2016 [1]. Interestingly, most cases occurred after 2010, while only one case was declared before 2010. Even though the methodology of this study was prone to declaration bias, it suggests that recent factors (i.e., changes in drug administration, modification of industrial process…) may account for the increasing incidence of AICN [1,13]. The higher incidence of AICN was also underlined in another French study that analyzed AKI in patients exposed to high dose AMX for the treatment of osteoarticular infections [2]. These reports, which are finally limited to the French territory, led the French National Drug Agency (ANSM) to conduct a nationwide study to analyze all cases of AICN declared to the French RPVs until 2016 [14]. In this report, 180 potential AICN cases following AMX treatment, either administered alone or in combination with clavulanate were identified, 82% of them diagnosed after 2010, mainly after IV administration. Although this inquiry does not allow to definitively confirm risk factors of AICN, it led to the following recommendations: limitation of AMX daily dose to 12 g, limitation of each administration to a maximum of 2 g to be infused in at least 20 min, favoring diuresis using IV hydration, and monitoring of renal function under treatment. In a recent report, most of these cases were reported in the English literature, showing older age and endocarditis as risks factors for AICN in medical patients [13].

Importantly, the diagnosis of AICN is far from being consensual and is usually retained after exclusion of other potential causes of AKI. The detection of AMX crystalluria allows to confirm the diagnosis, however, due to technical and logistical issues, it is rarely performed in practice. Of note, in the study of Vodovar et al., only 20% of AICN cases were screened for AMX crystals in urine [1].

As underlined above, given the declarative nature of RPV databases, AICN may be underestimated. Moreover, its real incidence remains to be determined, especially in patients receiving high AMX doses. Thus, the objective of the present study was to analyze the incidence of severe AKI Kidney Disease: Improving Global Outcome (KDIGO) 2 or 3) related to AICN by studying all consecutive patients treated with high dose AMX during a 3 year period in the University Hospital of Angers. The second objective was to describe AICN cases and identify potential risk factors in patients exposed to high AMX doses.

## 2. Materials and Methods

### 2.1. Selection of Patients

We conducted a retrospective study between 1 January 2015 and 31 December 2017. Patients admitted to the Angers University Hospital during the period and who received at least one dose of AMX were screened through the prescription software database (Crossway, Boulogne Billancourt, France). The software has been implemented in all hospital units since 2013, except in critical care medicine units. Adult patients that received at least 8 g/day of AMX using IV route for at least 24 h were included in the study. Patients that were admitted directly to critical care units or that received AMX for less than 24 h were excluded. The protocol was approved by the ethics committee of our institution (authorization number 2018/40).

### 2.2. Data Collection

For all patient included in the study, baseline characteristics, including demographic data and medical history were retrieved from medical records. Causes of admission (medical or surgical), infectious sites and identified germs, concomitant antibiotic use, and evolution of renal function from admission to discharge or death, daily dose, type of administration (discontinuous vs. continuous), duration of AMX treatment were also collected. Estimated glomerular filtration rate (GFR) was determined using the modified diet in renal disease (MDRD) formula. AKI was defined according to KDIGO classification based on creatinine variations, as urine outpout was not available for most patients [15].

Patients that developed AKI of at least KDIGO stage 2 or 3 were identified and their medical records were extensively analyzed. AKI characteristics, including urine sediment and proteinuria, kidney radiological exams (echography and computed tomography (CT) scan), temporality between AKI and initial admission, initiation of AMX, administration of other nephrotoxic drugs or contrast iodine injection, AMX withdrawal or dose modification, and recovery of AKI were collected.

### 2.3. Cases Definition

Medical records of patients that developed AKI of at least KDIGO stage 2 or 3 [15] were extensively analyzed and considered for AICN diagnosis. Medical records were reviewed by a nephrologist (JFA) and a specialist in pharmacovigilance (GD) in a blinded fashion. AICN diagnosis was retained in accordance with diagnosis criteria proposed by Vodovar et al. [1,13]. Vodovar criteria include inclusion and exclusion criteria. Briefly, to be classified as AICN, the following criteria were necessary: AKI, microscopic or macroscopic hematuria, and temporal relationship between AMX and AKI. AKI cases unrelated to drug administration were excluded, as were AKI that developed immediately after surgery, in the context of severe sepsis, dehydration, pancreatitis, heart failure, or rhabdomyolysis. We also excluded as potential AICN, AKI that developed along with administration of other nephrotoxic drugs (i.e., acyclovir or aminoglycosides) or after iodine contrast injection, even when AMX appeared to be most probable the causative agent. AKI associated with signs suggestive of drug induced acute interstitial nephritis were excluded (fever, rash, elevated liver enzymes, hypereosinophilia).

In case of discordant evaluation between investigators, medical records were analyzed and resolved by a second nephrologist (JFS).

### 2.4. Statistical Analyses

Continuous variables are presented as median and interquartile (IQR). Categorical variables are presented with their absolute value and percentage. Differences between groups were analyzed using the χ^2^ test (or Fisher exact test when applicable) for categorical variables and the Mann–Whitney U test for continuous variables. Univariate logistic regression analysis was used to study risk factors associated with AICN development. All the statistical tests were performed to the two-sided 0.05 level of significance. Statistical analysis was performed using SPSS software^®^ 23.0 for Macintosh and Graphpad Prism^®^ for Macintosh.

## 3. Results

### 3.1. Population Characteristics

Between 1 January 2015 and 31 December 2017, 165,162 patients were admitted to the University Hospital of Angers (excluding admissions to the intensive care units). Among them, 3754 (2.27%) received AMX orally or IV. Among patients that received IV AMX (*n* = 1303, 34.7%), 358 patients (27.5%) received doses equal or superior to 8 g for at least 24 h (Figure 1).

The population was predominantly composed of males (68.2%) with a median age of 71 years old. Half had hypertension and 22.9% had diabetes mellitus. The cause for admission was predominantly medical (78.5%) and the median length of stay was 20 days. The three most frequent causes of AMX administration were endocarditis, osteo-articular infection, and central nervous system infection. Most patients had a positive microbiological culture, and most frequently Gram-positive germs were identified. These data are reported in Table 1.

### 3.2. Amoxicillin Treatment

The median daily administered dose of AMX was 12 g/day, which represented a median dose of 152 mg/kg/day. Appendix A shows the distribution of AMX dose in the population. The median duration of AMX treatment was 14 days and the mean cumulated dose was 158 g per patient. AMX was rather administrated in a discontinuous manner as compared to continuous IV infusion. About half of patients received other antibiotics, concomitantly or not to AMX treatment, during their stay. The most used antibiotic classes were fluoroquinolones, aminoglycosides and cephalosporins. These data are summarized in Table 2.

### 3.3. Acute Kidney Injury and Mortality

We next analyzed renal function and identified patients that developed AKI episodes according to the KDIGO classification. Median eGFR on admission in the whole population was 91.9 (67.8–127.6) mL/min/1.73 m^2^. AKI was present in 12.6% of patients on admission and developed in 73 patients (20.4%) during hospital stay. KDIGO stage 2–3 AKI was diagnosed in 42 patients (11.7%), representing 56.9% of all AKI episodes. Nine patients (2.5%) required renal replacement therapy. None of the AKI patients underwent renal biopsy. These data are detailed in Table 3.

As compared to patients that did not develop AKI, patients with AKI had more frequently hypertension, diabetes mellitus, were more frequently admitted for a surgical reason, and had more frequently endocarditis. While serum creatinine and eGFR were comparable between groups on admission, patients with AKI had higher serum creatinine and lower eGFR at discharge. Hospital mortality was also significantly higher in patients that developed AKI. These results are reported in Appendix A.

### 3.4. AICN Cases

The 42 patients that developed KDIGO stage 2–3 AKI were extensively studied. Among these patients, case analyses concluded to a high probability of AICN in 16 patients (38.1%), representing an incidence of 4.5% (16/358) of patients exposed to high IV AMX doses over the three-year period of the study. In the remaining 26 patients, another AKI cause was retained: pre-renal AKI, predominantly post-operative AKI, was highly probable in 16 patients; concomitant nephrotoxic drugs were administered in 2 cases; obstructive AKI was retained in 2 cases; and other or unresolved AKI causes were observed in 6 cases.

We next analyzed the clinical and biological presentation of the 16 AICN cases (Table 4). AKI developed after a median delay of 5.5 days following AMX initiation. Median maximum serum creatinine was 322 µmol/L and 6 patients (37.5%) required hemodialysis. Almost all patients presented with microscopic hematuria, leukocyturia, and mild proteinuria. Macroscopic hematuria was observed in 5 patients (31.3%). Crystalluria was searched in 13 patients (81.3%) after a median delay of 5 days from AKI diagnosis and was positive in 2 patients (15.4%). In all but one patient, AMX withdrawn or dose was decreased after a median delay of 1 days following AKI diagnosis. Most patients (68.7%) recovered completely from AKI, but 5 patients (31.3%) had not yet recovered at hospital discharge.

As compared to patients with other AKI causes, females were significantly more represented in the AICN group (Appendix A). We did not observe any other significant difference between the AICN group and the other AKI group.

Interestingly, among KDIGO stage 2–3 AKI, 7 cases of AICN were suspected at the time of hospitalization and declared to the RPV, 6 of them were classified as AICN cases in the present study, representing a declaration rate of 37.5%.

### 3.5. Factors Associated with AICN Development

In a last step, we analyzed factors associated with the risk of developing AICN. Univariate analysis (Table 5), performed after exclusion of patients with KDIGO stage I AKI and of patients with other AKI causes, showed that female gender was the only factor significantly associated with an increased risk of AICN (OR 2.93, *p* = 0.039). Other factors and, especially AMX dose, even when indexed to weight and impaired GFR at baseline, were not associated with the risk of developing AICN in our study.

## 4. Discussion

This work is best to our knowledge the first that aimed to evaluate the incidence of AICN in patients treated with high IV AMX doses. The main results of this study is the observation of a 4.5% AICN incidence over a 3 years period in patients receiving high IV doses of AMX, and that AICN represented the cause of AKI in more than one third of KDIGO stage 2–3 AKI in this population. Thus, as compared to previous reports [1,13], these results suggest that AMX nephrotoxicity may be underestimated. Interestingly, only female gender appeared as a risk factor of AICN, while AMX dose (even if indexed to weight) and modalities of AMX administration (continuous vs. discontinuous infusion) were not significantly associated with AICN.

Even though AICN has been described in the literature since the 1990s, published data rely mainly on case reports or cases series [9,10,11,12,16,17] and on pooled analyses of cases reported to the RPVs [1]. Despite these previous studies do not allow to conclude on AICN incidence and to identify risk factors, AICN appeared related to high AMX doses administered by IV route, usually above 8 g/day. Thus, we decided to evaluate AICN incidence in such patients which are those with the higher AICN risk. It is important to underline that there is no consensual definition of AICN. In previous studies, the diagnosis of AICN was retained after exclusion of other AKI causes, in temporally compatible cases according to AMX administration and after exclusion of patients with concomitant administration of other nephrotoxic drugs [1]. In these conditions, the detection of birefringent needle-shaped crystals in fresh urines is very suggestive of AICN [4,12]. However, the search for crystalluria is rarely performed or is performed but often uninterpretable due to technical issues. In our study, crystalluria was searched in 13/16 (81.3%) AICN cases but crystals were identified in only 2 patients (15.4%). The low incidence of crystal detection in our study may be explained by the long delay between AKI onset and urinalyses, which was 4.5 days. Moreover, we assume that the technical conditions to identify urine crystals were not present (fresh urines, analysis within 2 h). In comparison, in the study of Vodovar et al., among the 45 patients diagnosed with AICN, a crystal search was performed in 20% of patients [1], and detected in 68/180 AICN patients (38%) in the ANSM nationwide study [14]. In order to limit AKI misclassification in our study, medical files of patients with AKI stage 2–3 were analyzed by 2 practitioners, including by a specialist in pharmacovigilance, and classified according to Vodovar AICN criteria, which were used in past studies [1,13].

The clinical presentation of patients at AICN onset in our study was globally in accordance with previous reports. AICN has been reported in patients who received AMX for prophylaxis mainly before surgery or to treat established infections [1,13]. Given the design of the present study which included patients with AMX treatment for at least 24 h, our analysis mainly focused on patients that received AMX for treatment of ongoing infection and not for those where AMX was used for prophylaxis. We observed that AICN onset occurred after a median delay of 5.5 days, which is in accordance with the delay observed in the ANSM nationwide study (7 days) in patients receiving AMX for ongoing infection [14] and in the most recent report of Vodovar [13]. Macroscopic hematuria was observed in 31.3% of AICN patients, and microscopic hematuria and leukocyturia were detected in all patients. These observations are unspecific, but were also observed in previous reports.

All but one AICN patient had AMX discontinuation or dose reduction, which suggests that AICN was suspected in almost all patients at the time of hospitalization. However, and importantly, only 6/16 AICN cases (37.5%) were declared by clinicians to the RPV, which represents a low declaration rate given that most cases were suspected to be related to AMX administration by clinicians [18,19].

Most previous reports underline that AICN is usually a reversible complication [1,2,14]. In the ANSM nationwide study, dialysis was required in 23% of cases and mortality rate was 2% [14]. In the present study, 3 patients (18.8%) of the AICN group died during initial admission and kidney function did not return to baseline at hospital discharge in 5 patients (31.3%). Moreover, 6 patients (37.5%) needed hemodialysis transiently. The poor prognosis observed in our study may be related to our case definition which focused on patients with more severe AKI (KDIGO 2–3), and also to our inclusion criteria that selected patients with severe infections. However, it also suggests that AICN may not be a complication as so benign as previously thought. Even if the present study does not allow to analyze the relationship between AICN and mortality, AKI has been found constantly associated with the risk of both in-hospital and long-term mortality in a wide range of medical and surgical conditions [20,21]. Moreover, several recent reports also showed that AKI episodes are associated with an increased risk of developing chronic kidney disease in the future [22,23]. Thus, caution should be taken with the “benign” nature of AICN.

We observed that female gender was the sole risk factor of AICN in our study, while males represented 80.8% of other AKI causes unrelated to AICN. Interestingly, in previous studies, females appeared over-represented as compared to males [1,2,14] in AICN, while male gender is classically over represented in AKI. We did not identify any other risk factor for AICN in our population. Notably, nor body mass index, nor AMX dose, which were suggested to be risk factors in previous studies [1,2,14], were associated with AICN risk in our work. An explanation for this may be that we selected patients with high AMX dose as part of the inclusion criteria of the study, and that finally, most patients were in the upper limit range of AMX dose administration, not allowing to identify a dose relationship. We have no definite explanation to the reason why female gender may expose to a higher risk of AICN. However, considering that AMX urine concentration is a major determinant of AICN risk [1], we can suggest the following hypothesis. As compared to males, females have a lower number of nephrons, a difference which is in part genetically determined [24]. Moreover, in respect to body compartments, females have a higher fat mass proportion as compared to males [25]. Thus, the AMX dose reported to lean mass may be higher in females as compared to males, and given the lower nephron number in females, the AMX urine concentration may be higher for an equivalent administered dose.

Our study has several limitations, starting with its retrospective design. As underlined previously, another limitation is the definition of AICN cases that relies on temporal causality analysis and exclusion of differential diagnosis, based on data retrieved from medical records. In this view, the methodology used in the present study is comparable, and even more restrictive, than those used in previous works and may have rather resulted in an underestimation of AICN incidence given that some potential AICN cases were excluded because of concomitant administration of other nephrotoxic drugs. Moreover, we did not analyze the imputability of AMX in patients with mild AKI (KDIGO stage 1 AKI), which may also have underestimated AICN incidence. Another limitation of our study is that we excluded patients initially admitted to ICU.

Studies from the French RVPs showed that AICN incidence increased dramatically after 2010 [2,14]. The French nationwide study conducted by the ANSM did not allow to identify factors that could explain the increased incidence of AICN [14]. This discussion is beyond what we can infer from the present work, however, we can suggest that some changes in clinical practice may account for a more frequent use of high AMX doses. Endocarditis was the first indication of high dose AMX use in the present study and also in previous ones. This may be the consequence of guidelines recommending its use as a first line antibiotic treatment after 2009 [26,27].

## 5. Conclusions

The present study shows that the incidence of AICN may be underestimated in patients treated with high doses AMX. Moreover, our data suggest that AICN prognosis may be less favorable than previously thought. Thus, we recommend a close monitoring of renal function of patients exposed to high dose AMX and to consider an early switch to another antibiotic class if severe AKI develops under AMX treatment.

## Figures and Tables

**Figure 1 jcm-09-02022-f001:**
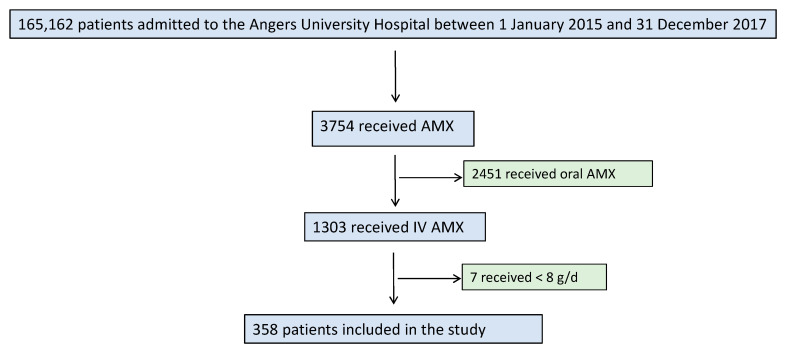
Flowchart of the study.

**Table 1 jcm-09-02022-t001:** Characteristics of the population. Results are presented as median (IQR) and absolute value (percentage). The results are presented as median (IQR) for continuous variables and number (%) for categorical variables.

Number of Patients	358
Gender, M/F (%)	244/114 (68.2/31.8)
Age, years	71.0 (60–82)
Weight, Kg	75.0 (64.1–88.0)
BMI, Kg/m^2^	26.0 (22.8–30.6)
Hypertension, *n* (%)	186 (52.0)
Diabetes mellitus, *n* (%)	82 (22.9)
Length of hospital stay, days	20.0 (13–31.8)
Type of admission, *n* (%)	
Medical	281 (78.5)
Surgical	77 (21.5)
Nature infectious event, *n* (%)	
Endocarditis	110 (30.7)
Osteo-articular	68 (19.0)
Septicemia	61 (17)
Central nervous system	31 (8.7)
Lung	14 (3.9)
Urinary tract	4 (1.1)
Digestive	2 (0.6)
Other	68 (19.0)
Microbiology, *n* (%)	
Patients with at least a positive bacterial culture	310 (86.6)
Type of bacteria	
Gram positive	280 (90.3)
Gram negative	30 (9.7)

**Table 2 jcm-09-02022-t002:** Amoxicillin treatment and associated antibiotics. Results are presented as median (IQR) and absolute value (percentage). * (min–max).

Amoxicillin dose, g/24 h	12 (8–18) *
Amoxicillin dose, mg/kg/24 h	152.0 (123–178)
Cumulated dose, g	158.0 (72–336)
Treatment duration, days	14.0 (7–28)
Type of IV administration, *n* (%)	
Continuous infusion	120 (33.5)
Discontinuous infusion	238 (66.5)
Other antibiotics, *n* (%)	198 (55.3)
Fluoroquinolones	66 (33.3)
Aminoglycosides	50 (25.3)
Céphalosporins	38 (19.2)
Glycopeptides	7 (3.5)
Others	37 (18.7)
Acyclovir, *n* (%)	10 (2.8)

**Table 3 jcm-09-02022-t003:** Kidney function and acute kidney injury (AKI) characteristics in the study population. Results are presented as median (IQR) and absolute value (percentage).

	All, *n* = 358
Serum creatinine at admission, µmol/L	77.0 (60.0–100.8)
GFR at admission, mL/min/1.73 m^2^	91.9 (67.8–127.6)
Lowest serum creatinine, µmol/L	65.0 (53.0–82.0)
Highest serum creatinine, µmol/L	88.0 (69.0–130.5)
AKI at admission, *n* (%)	45 (12.6)
AKI during hospital stay, *n* (%)	73 (20.4)
KDIGO stage 1	31 (8.7)
KDIGO stage 2	16 (4.5)
KDIGO stage 3	26 (7.3)
Need for renal replacement therapy, *n* (%)	9 (2.5)
Serum creatinine at discharge, µmol/L	72.5 (61.8–95.3)
GFR at discharge, mL/min/1.73 m^2^	90.2 (65.7–114.1)

**Table 4 jcm-09-02022-t004:** Description of AMX-induced crystal nephropathy (AICN) cases. Results are presented as median (IQR) and absolute value (percentage).

	*n* = 16
Delay between AMX initiation and AKI, days	5.5 (4.0–10.0)
Maximum serum creatinine, µmol/L	322.0 (262.5–462.0)
Evolution of AKI	
Complete AKI recovery *, *n* (%)	11 (68.7)
Delay to recovery, days	16.0 (8.0–35.0)
Non recovery *, *n* (%)	5 (31.3)
Need for hemodialysis, *n* (%)	6 (37.5)
Number of hemodialysis sessions, *n* (min-max)	2.0 (1–3)
AMX treatment arrest or dose reduction	15 (93.8)
Delay from AKI diagnosis, days (min-max)	1 (1–22)
AMX arrest, *n* (%)	11 (68.8)
AMX dose reduction, *n* (%)	4 (25.0)
Biological and radiological evaluation, *n* (%)	
Kidney ultrasound	15 (93.8)
Urine cytology	11 (68.7)
Proteinuria quantification	13 (81.3)
Search for AMX crystalluria	13 (81.3)
Delay between AKI and AMX crystal search, days	5 (2–6.5)
AKI presentation	
Microscopic hematuria **, *n* (%)	11 (100)
Macroscopic hematuria, *n* (%)	5 (31.3)
Leukocyturia **, *n* (%)	11 (100)
Proteinuria ***, g/g creatininuria	0.68 (0.32–1.46)
Urine crystals **, *n* (%)	2 (15.4)

* at the end of hospitalization or before death; ** Among screened patients; *** Among screened patients without macroscopic hematuria.

**Table 5 jcm-09-02022-t005:** Univariate analysis for risk factors associated with AICN. Patients with non AICN AKI as well as patient with Kidney Disease:Improving Global Outcome (KDIGO) stage I AKI were excluded.

	Univariate Analysis	
	OR (CI)	*p*
Age (per year)	1.03 (0.99–1.07)	0.098
Gender (female)	2.93 (1.06–8.11)	0.039
Hypertension (yes)	2.28 (0.77–6.73)	0.136
Diabetes mellitus (yes)	1.74 (0.58–5.21)	0.321
BMI >30 kg/m^2^	0.85 (0.22–3.19)	0.804
Medical admission (versus surgical)	1.60 (0.35–7.25)	0.543
Endocarditis	1.56 (0.55–4.45)	0.401
Amoxicillin dose (g/day)	1.10 (0.83–1.45)	0.508
Amoxicillin dose (>12 kg/day)	1.63 (0.45–5.89)	0.454
Amoxicillin dose (>200 mg/Kg/day)	2.63 (0.70–9.90)	0.153
Type amoxicillin administration (discontinuous)	1.44 (0.52–3.98)	0.493
eGFR at admission, mL/min/1.73 m^2^	0.99 (0.98–1.00)	0.133
eGFR <60 mL/min/1.73 m^2^ at admission	2.16 (0.76–6.21)	0.150

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
