# Peer review of "High Incidence of Amoxicillin-Induced Crystal Nephropathy in Patients Receiving High Dose of Intravenous Amoxicillin"

_jcm, 2020, doi:10.3390/jcm9072022_

Round 1

Reviewer 1 Report

To the Authors

In this original article, Anne-Sophie Garnier et al. are presenting a retrospective analysis on Amoxicillin Induced Crystal Nephropathy (AICN) in a cohort of patients admitted to the Angers University Hospital between 2015 and 2017.

French pharmacovigilance registries documented and increase of the incidence of AICN during the years (especially from 2010).

Anyway, given the declarative nature of those registries, the true incidence of AICN might be underestimated.

In this present work the Authors performed a retrospective analysis on patients admitted to the hospital who subsequently developed acute kidney injury related to AICN. The primary objective was the evaluation of AICN incidence and the secondary objective to identify potential risk factors.

The study meets the primary objective, estimating an incidence of 4.5% of patients undergoing AICN after amoxicillin exposure, and recognizes the female sex as a possible risk factor.

In my opinion the study is conducted in an appropriate way, considering also the double blinded control to detect AICN cases from the medical records.

The description of the study population is accurate and to my opinion selection biases are not present.

I have just a few minor concerns:

Lines 91-92 – a possible explanation of the incidence increase of AICN in the pharmacovigilance registry is given. Do you have references that can sustain these ideas? In case I would add them.

Lines 103-104 – also here I think some evidences from the literature should be properly cited.

Finally, I would recommend a thorough language check since a few grammar and syntactic errors are present.

Author Response

We acknowledge Reviewer 1 for his valuable comments.
To the Authors
In this original article, Anne-Sophie Garnier et al. are presenting a retrospective analysis on Amoxicillin Induced Crystal Nephropathy (AICN) in a cohort of patients admitted to the Angers University Hospital between 2015 and 2017.
French pharmacovigilance registries documented and increase of the incidence of AICN during the years (especially from 2010).
Anyway, given the declarative nature of those registries, the true incidence of AICN might be underestimated.
In this present work the Authors performed a retrospective analysis on patients admitted to the hospital who subsequently developed acute kidney injury related to AICN. The primary objective was the evaluation of AICN incidence and the secondary objective to identify potential risk factors.
The study meets the primary objective, estimating an incidence of 4.5% of patients undergoing AICN after amoxicillin exposure, and recognizes the female sex as a possible risk factor.
In my opinion the study is conducted in an appropriate way, considering also the double blinded control to detect AICN cases from the medical records.
The description of the study population is accurate and to my opinion selection biases are not present.
I have just a few minor concerns:

We acknowledge Reviewer 1 for these comments.

Lines 91-92 – a possible explanation of the incidence increase of AICN in the pharmacovigilance registry is given. Do you have references that can sustain these ideas? In case I would add them.
We agree with Reviewer 1. In fact, data from French pharmacovigilance were reported in two recent studies: ref 1 and ref 13. We added these references to line 91-92.
Lines 103-104 – also here I think some evidences from the literature should be properly cited.
We agree with reviewer 1 and we refer here to reference 13 which we added.
Finally, I would recommend a thorough language check since a few grammar and syntactic errors are present.
We revised the manuscript and corrected for the grammar.

Reviewer 2 Report

Garnier et al. conducted retrospective study to determine the incidence of amoxicillin (AMX)-induced crystal nephropathy and to identify potential risk factors in patients exposed to high AMX doses.

General comments

The results of this study are very important, since it tried to clarify the incidence of Amoxicillin-Induced Crystal Nephropathy (AICN) and its risk factors. This information assists physicians to avoid AICN and to improve renal prognosis of patients with high dose of amoxicillin treatment. Authors should respond to the next comments.

Major comments

  • In Page 12, line 4, Please explain the reason more clearly, why authors declared that AMX nephrotoxicity may be underestimated. Have there been any data existed being below a 4.5% incidence?

Please note that the incidence AICN generated from this study is not annual incidence, but three-year incidence.

Minor comments

  • In page 5, line 19, ‘to be classify’ should be corrected to ‘ to be classified’.

  • In page 12, line 14, ‘ AMX’ should be replaced by ‘AICN’.

  • In page 14, line 13, ‘quantity’ should be replaced by ‘number’.

  • In Supplemental Figure 1, Y axis, ‘nomber of patients’ should be corrected to ‘number of patients’.

Author Response

We acknowledge Reviewer 2 for his valuable comments.

Garnier et al. conducted retrospective study to determine the incidence of amoxicillin (AMX)-induced crystal nephropathy and to identify potential risk factors in patients exposed to high AMX doses.

General comments

The results of this study are very important, since it tried to clarify the incidence of Amoxicillin-Induced Crystal Nephropathy (AICN) and its risk factors. This information assists physicians to avoid AICN and to improve renal prognosis of patients with high dose of amoxicillin treatment. Authors should respond to the next comments.

Major comments

  • In Page 12, line 4, Please explain the reason more clearly, why authors declared that AMX nephrotoxicity may be underestimated. Have there been any data existed being below a 4.5% incidence?

We acknowledge Reviewer 2 for this comment. We agree that this sentence needs to be clarified.

In fact, based on French pharmacovigilance reports, and the few AICN cases reported as case reports or case series in the literature, AICN incidence is expected to be much lower. However, our study is best to our knowledge the first to analyze AICN incidence in a population of consecutive patients exposed to high IV AMX doses.     

Moreover, given that AICN was not considered in patients with mild AKI (stage 1) in our study, we can hypothesize that some stage 1 KDIGO were related to AICN, thus underestimating also AICN incidence in the whole cohort.

We modified the text as follows to clarify:

“Thus, as compared to previous reports (1,15), these results suggest that AMX nephrotoxicity may be underestimated

Please note that the incidence AICN generated from this study is not annual incidence, but three-year incidence.

We totally agree with reviewer 2 and added this precision :

“The main results of this study is the observation of a 4.5% AICN incidence over a 3 years period in patients receiving high IV doses of AMX..”

Minor comments

  • In page 5, line 19, ‘to be classify’ should be corrected to ‘ to be classified’.

We corrected this mistake.

  • In page 12, line 14, ‘ AMX’ should be replaced by ‘AICN’.

This was corrected.

  • In page 14, line 13, ‘quantity’ should be replaced by ‘number’.

This was corrected

  • In Supplemental Figure 1, Y axis, ‘nomber of patients’ should be corrected to ‘number of patients’.

We corrected this error.

Reviewer 3 Report

The authors have found that high IV doses of amoxicillin results in high incidence of crystal nephropathy. I am wondering why that is not observed following oral dosing considering its high oral bioavailability of 77%. Please clarify.

Author Response

We acknowledge Reviewer 3 for having reviewed our work.

The authors have found that high IV doses of amoxicillin results in high incidence of crystal nephropathy. I am wondering why that is not observed following oral dosing considering its high oral bioavailability of 77%. Please clarify.

We acknowledge reviewer 3 for this important comment. In fact, AICN has been reported with oral AMX use. However, its incidence with oral administration seems to be much lower as compared to IV administration, probably because of full availability and also because high doses, usually given in severe infections, are most frequently administered intravenously.

We added the following precision to the introductive part: page 4 line 99: “In this report, 180 potential AICN cases following AMX treatment, either administered alone either in combination with clavulanate were identified, 82% of them diagnosed after 2010, mainly after IV administration”
